# Real-World Clinical Outcomes and Replacement Factor VIII Consumption in Patients with Haemophilia A in Italy: A Comparison between Prophylaxis Pre and Post Octocog Alfa (BAY 81-8973)

**DOI:** 10.3390/jcm11123434

**Published:** 2022-06-15

**Authors:** Paolo Angelo Cortesi, Giovanni Di Minno, Ezio Zanon, Gaetano Giuffrida, Rita Carlotta Santoro, Renato Marino, Lucia Sara D’Angiolella, Ippazio Cosimo Antonazzo, Ginevra Squassabia, Francesco Clemente, Danilo Di Laura, Ernesto Cimino, Samantha Pasca, Daniela Nicolosi, Lorenzo Giovanni Mantovani

**Affiliations:** 1Research Centre on Public Health (CESP), University of Milano-Bicocca, 20900 Monza, Italy; lucia.dangiolella@gmail.com (L.S.D.); ippazio.antonazzo@unimib.it (I.C.A.); g.squassabia@campus.unimib.it (G.S.); f.clemente4@campus.unimib.it (F.C.); dilaura.danilo@gmail.com (D.D.L.); lorenzo.mantovani@unimib.it (L.G.M.); 2Value-Based Healthcare Unit, IRCCS Multimedica, 20099 Sesto San Giovanni, Italy; 3Regional Reference Center for Coagulation Disorders, Federico II University Hospital, 80131 Naples, Italy; diminno@unina.it (G.D.M.); ernesto.cimino@unina.it (E.C.); 4Hemophilia Center, University Hospital of Padua, 35128 Padua, Italy; zanezio61@gmail.com (E.Z.); sampasca27@gmail.com (S.P.); 5Division of Haematology, A.O.U. Policlinico Vittorio Emanuele, 95123 Catania, Italy; gaegiuffrida@gmail.com (G.G.); danielanicolosi03@gmail.com (D.N.); 6Centre for Haemorrhagic and Thrombotic Disorders, “Pugliese Ciaccio” Hospital, 88100 Catanzaro, Italy; ritacarlottasantoro@gmail.com; 7Haemophilia and Thrombosis Center, Giovanni XXIII Hospital of Bari, 70126 Bari, Italy; remarino64@gmail.com

**Keywords:** haemophilia A, treatment burden, cost-effectiveness, infusion frequency

## Abstract

(1) Background: new generations of rFVIII products offered the possibility to improve personalized therapeutic approaches, reducing the number of infusions or increasing the protection against bleeding risk. The aim of this study was to assess the effectiveness of prophylaxis with BAY 81-8973 (octocog alfa, Kovaltry^®^, Bayer Pharma AG) in the real-world setting and its impact on FVIII consumption compared to previous standard half-life treatments. (2) Methods: a retrospective observational study was conducted in five Italian Haemophilia Centers. Patients with haemophilia A under prophylactic treatment with BAY 81-8973 for at least one year, and previously on prophylaxis with a different product were included in the study. Annual bleeding rate (ABR) and annual FVIII consumption were compared. (3) Results: forty-four patients were included in the study. After switching to BAY 81-8973, ABR was significantly reduced (1.76 vs. 0.23; *p* = 0.015), the percentage of patients with zero bleeds increased from 54.6% to 84.1% (*p* = 0.003), and the overall FVIII consumption decreased by 25,542 (−7.2%, *p* = 0.046) IU per patient-year. Patients treated every 3 days or 2 times per week increased from 0% to 27.3%. (4) Conclusion: our results suggest that prophylaxis with BAY 81-8973 can improve clinical outcomes and reduce FVIII consumption, in the real-world practice, compared with the previous prophylaxis regimen with standard half-life products.

## 1. Introduction

Haemophilia is a rare congenital bleeding disorder characterized by gene abnormalities leading to defective or missing clotting Factor VIII (FVIII), called haemophilia A (HA), and Factor IX (FIX), called haemophilia B (HB) [1,2]. The prevalence of HA is approximately five times that of HB, with a worldwide frequency estimated at one per 5000–7000 male births [1,2]. Symptoms are characterized by bleeding episodes that occur mainly in the joints (haemarthroses) and muscles (haematomas), with frequency and severity proportional to the extent of the coagulation defect and which can be spontaneous and post-traumatic [3]. In particular, recurrent haemarthrosis leads to serious deterioration of joint structures, with a consequent reduction in function and atrophy of the associated skeletal muscles, with a significant impact on patents’ quality of life [1]. Haemophilia patients can also experience rare brain bleeds, both intracranial and extracranial; these are the most severe and life-threatening complication in the neonatal period [1]. In the last decades, thanks to the availability of recombinant FVIII (rFVIII) concentrates characterized by high safety and efficacy, and the spread of continuous treatment regimens aimed at prevention of bleeds (prophylaxis), there has been a continuous improvement in the treatment of patients with haemophilia.

Scientific evidence has shown that, compared with on-demand treatment, prophylaxis is associated with better clinical outcomes, including joint status and quality of life [1,2,3,4]. Nevertheless, the prophylaxis offered to the patients is not yet completely optimal and open problems remain for both patients and clinicians related to current therapies as the burden (number of infusion) associated with the treatment and the possible through level achievable, and the weight-based fixed-dose approach frequently used in the past [5,6].

To overcome these difficulties, clinicians moved towards individualized treatment approaches that are defined on the basis of the characteristics of each patient, their lifestyle, bleeding phenotype, preferences, and those of the individual product, including pharmacokinetic properties [7,8]. The possibility of a personalized approach was significantly improved also thanks to the advance in treatment options, as rFVIII products offer the possibility of a lower frequency of infusions compared with the usual every-other-day or three-times-per-week regimen [4,9,10]. One of the first rFVIII concentrates available on market that allowed a two-times-a-week prophylaxis regimen was BAY 81-8973 (octocog alfa, Kovaltry^®^, Bayer Pharma AG, Berlin, Germany), an unmodified, full-length, standard half-life (SHL) rFVIII concentrate approved in 2016 for on-demand treatment, prophylaxis, and perioperative management of patients with haemophilia A. Even if BAY 81-8973 is not an extended half-life product, it is the first rFVIII with a label of two times a week prophylaxis regimen that can be used in some patients. Understanding the impact of these products on the effectiveness and costs in clinical practice has become particularly important to help healthcare decision makers in understanding the real value of these innovations [11].

So far, the possible impact of BAY 81-8973 has been assessed with indirect comparisons based on clinical trials results and consumptions as reported in summary of product characteristics [9,12] and only preliminary results of one study on real-world prophylactic treatment with unmodified full-length recombinant FVIII BAY 81-8973 are available [13]. The aim of this study was to fill this gap, assessing the effectiveness of prophylaxis with BAY 81-8973 in the real-world setting and its impact on FVIII consumption compared with a previous treatment with SHL recombinant product.

## 2. Materials and Methods

Five Italian Haemophilia Centers participated in a retrospective, observational, multicenter cohort study: (1) Regional Reference Center for Coagulation Disorders, Federico II University Hospital, Naples, (2) Hemophilia Center, University Hospital of Padua, Padua (3) Division of Haematology, A.O.U. Policlinico Vittorio Emanuele, Catania, (4) Center for Haemorrhagic and Thrombotic Disorders, “Pugliese Ciaccio” Hospital, Catanzaro, and (5) Haemophilia and Thrombosis Center, Giovanni XXIII Hospital of Bari, Bari.

The study protocol and the informed consent form were approved by Institutional Ethics Committees of all participating Centers. The study protocol is in agreement with the principles established by the 18th World Medical Assembly (Helsinki, 1964).

All patients with haemophilia A without inhibitors on prophylaxis with BAY 81-8973 were included, regardless of age and severity of disease. Patients were routinely managed according to Italian and European guidelines, and the approved clinical protocols of each center. Data inclusion in this study had no impact on the intended management, which was determined according to the criteria of the treating specialist.

### 2.1. Inclusion and Exclusion Criteria

The study included subjects with the following characteristics: (a) diagnosis of severe, moderate, or mild congenital haemophilia A, (b) absence of inhibitors, (c) treated on prophylaxis with BAY 81-8973 for at least 12 months, (d) treated on prophylaxis with a different SHL rFVIII product in the 12 months prior to switch to BAY 81-8973, and (e) able to understand and sign the informed consent (parents for minor patients).

Consistently with the inclusion criteria, the study excluded subjects with: (a) bleeding disorders other than haemophilia A or acquired haemophilia A, (b) presence of inhibitor at the time of switch to BAY 81-8973, (c) use of on-demand regimen during the study period, (d) presence of any condition for which BAY 81-8973 was contraindicated, including hypersensitivity to the active ingredient or to any of the excipients contained in the drugs, and (e) subjects or parents NOT able to understand and sign the informed consent.

### 2.2. Data Collection

Patients that satisfied the inclusion and exclusion criteria were enrolled by haemophilic centers involved in the study between January 2019 and April 2020. Variables were retrospectively collected and recorded from the clinical history reported in the medical records. These included age, sex, haemophilia severity (severe, moderate or mild), time from the start of prophylaxis, and presence of target joints. In the 12 months before switching to BAY 81-8973 we collected information on annual bleeding rate (ABR), prescribed product and prophylaxis regimens (dosage and frequency of administration), annual FVIII consumption with respect to prophylaxis and treatment of bleeds, and reason for switch to BAY 81-8973. In the 12 months after switching to BAY 81-8973 we collected information on the ABR, prophylaxis regimens (dosage and frequency of administration), annual FVIII consumption for prophylaxis, and treatment of bleeds. The bleeding events were mainly recorded in the centers based on patients dairy or based on the events reported by the patients during the visits. Target joints were defined according to the World Federation of Hemophilia 2012 guidelines (https://elearning.wfh.org/resource/treatment-guidelines/, accessed on 3 March 2022).

### 2.3. Sample Size

Due to the observational characteristics of the study, all patients who satisfied the inclusion and exclusion criteria were included in the study.

However, the study was design to compare ABR from matched pairs of study subjects. Prior data indicate a difference in ABR between 1.4 and 2.2 and a standard deviation between 0.6 and 1.4 in observational pre-post study [14]. If the true difference in the mean response of matched pairs is 1.4 and the standard deviation is 2.5, we will need to study 36 pairs of subjects to be able to reject the null hypothesis that this response difference is zero with probability (power) 0.9. The Type I error probability associated with this test of this null hypothesis is 0.05.

### 2.4. Statistical Analysis

Descriptive analyses were conducted to describe the baseline characteristics of enrolled patients and summarize their clinical outcomes (ABR and proportion of patients with zero bleeds) and FVIII utilization, both during the prophylaxis with BAY 81-8973 and the previous product.

Comparison of clinical outcomes and FVIII consumption was performed with the Wilcoxon Signed-Rank Test for paired samples or Student *t* test for paired samples, after confirming the criteria for using parametric or nonparametric methods.

A *p*-value < 0.05 was considered to be statistically significant. All analyses were performed using SAS software (version 9.4; SAS Institute Inc., Cary, NC, USA).

## 3. Results

The study enrolled 44 patients with a mean (±SD) age of 37.2 (±17.4) years. All patients were male. Baseline demographic and clinical characteristics are shown in Table 1. Severe haemophilia A represented 95.5% (42/44) of patients enrolled. The median (range) time spent on a prophylaxis regimen before the switch to BAY 81-8973 was 9.0 (1–20) years, with the majority of patients (41/44, 93.2%) using a second generation rFVIII concentrate during the last year (Table 1). The main reasons for switching to BAY 81-8973 were the need for more protection against bleeds, and a better adherence to treatment thought a regimen with a fewer intravenous injections (Figure 1).

### 3.1. Clinical Outcomes

BAY 81-8973 prophylaxis reported a higher percentage of patients with zero bleeds compared with the previous prophylaxis (54.5% vs. 84.1%, *p* = 0.003) (Table 2). Mean ABR was significantly lower with BAY 81-8973 comparing the period pre- and post-switch (1.76 vs. 0.23; Δ = −1.53; *p* = 0.015). Similarly, among patients reporting at least one bleeding episode in both treatment periods, the mean ABR reported in the year before the switch to BAY 81-8973 was significantly higher compared with that observed during the prophylaxis with BAY 81-8973 (3.90 vs. 1.43; Δ = −2.47; *p* < 0.001) (Table 2). Either before or after the change of product, most breakthrough bleeds were joint bleeds.

### 3.2. Prophylaxis Regimen and FVIII Consumption

As shown in Table 3, prophylaxis with BAY 81-8973 was associated with a significant reduction in the mean total annual FVIII consumption compared with that performed in the previous year (−25,541.71 IU, *p* = 0.046). This reduction was mainly related to the lower ABR reported with BAY 81-8973. In fact, while reduction in the mean annual FVIII consumption for on-demand treatment of bleeds was 86.8% (1817.47 vs. 13,746.29 IU, Δ = −11,928.82 IU), reduction for prophylaxis was 4.0% (327,706.47 vs. 341,319.36 IU, Δ = −13,612.89 IU). Prophylaxis with BAY 81-8973 was associated with a higher dosage in terms of IU/kg administered per each infusion (+0.99 IU/kg, +3.2%) and a lower number of infusions per week (−0.21 infusions, −6.9%). The distributions of the infusion frequency in the period pre- and post-switch to BAY 81-8973 are showed in Figure 1. Relevant changes, in terms of number of patients, were observed mainly in the every-other-day regimen (from 13 to 4 patients), in the every-3-days (from 0 to 7), and in the 2-times-per-week (from 0 to 5) regimens. Overall, the percentage of patients treated every 3 days or 2 times per week increased from 0% to 27.3% using BAY 81-8973 (Figure 2).

## 4. Discussion

The possibility of personalized therapeutic approaches in haemophilia care has been significantly improved also thanks to the advance in treatment options, such as recent rFVIII products that allow a lower frequency of infusions compared with the usual every other day or three-times-per-week regimen of older standard half-life products [4,5,6,7,8,9]. Understanding the impact of these products on the effectiveness and costs in clinical practice has become particularly important to help healthcare decision makers in understanding the real value of these innovations [10]. To our knowledge, this is the first Italian study assessing the real-world impact of BAY 81-8973, one of the first rFVIII products to offer patients the possibility to be treated even with two administrations per week prophylaxis.

Our study enrolled 44 patients treated at five Italian Haemophilia Treatment Centers that switched their prophylaxis from a 2nd- or 3rd-generation SHL rFVIII product to BAY 81-8973, showing an improvement in clinical outcomes, dosing regimen, and drug consumption. These patients switched to BAY 81-8973 mainly to obtain more protection against bleeds through higher plasma factor concentrations and increase adherence to treatment by reducing the frequency of infusions. When treated with BAY 81-8973, patients reported a higher percentage of patients with zero bleeds (54.6% vs. 84.1%, *p* = 0.003) and a significantly lower ABR (1.76 vs. 0.23; *p* = 0.015). The clinical improvement was associated with a significant reduction of the total annual FVIII consumption, due to a slightly lower IU consumption for the prophylaxis regimen and a more significant reduction in IU consumption to manage bleeding events. These outcomes were obtained with an improvement of treatment burden. More than 25% of patients reported the every-3-days or 2-times-per-week regimens when treated with BAY 81-8973 compared with 0% when treated with the previous product. This result confirms what observed in the LEOPOLD studies program, i.e., BAY 81-8973 is able to provide effective prophylaxis with two injections per week in approximately 30% of patients.

Only preliminary results on MulTinational phAse IV study evaluating “Real-world” treatment patterns in previously treated haemophilia A patients Receiving BAY 81-8973 (octocog alfa) for routine prophylaxis (TAURUS), an international open-label, prospective, non-interventional, single-arm study are available to assess the real world improvement of these parameters in patients who switched from a different rFVIII product to BAY 81-8973, a non-modified, full-length, SHL rFVIII authorized for prophylaxis regimes up to 2 times a week [13].

The preliminary results of TAURUS study showed good levels of treatment satisfaction and adherence. TAURUS demonstrated a favorable PK profile of BAY 81-8973 in comparison with other standard half-life rFVIIIs. However, it should be noted that at the interim cut-off, most patients had not yet reached one year of observation, and the animalization of bleeds reported in a shorter time period results in less reliable annualized bleeding rate (ABR) estimates. Therefore, the median (Q1–Q3) number of actual total bleeds without animalization was provided in the paper: 2.0 (0.0; 5.0).

Our study results and preliminary results of TAURUS study went in the same direction confirming the positive impact of BAY 81-8973 prophylaxis and the possible advantage compared with 2nd- or 3rd-generation SHL-rFVIII. Further, the study provided complementary data affording a complete picture of the possible impact on BAY 81-8973 compared with previous SHL-rFVIII in terms of bleeding rate, treatment burden, patients satisfaction, FVII IU consumption, and PK profile. These improvements must be related mainly to the possibility of using more flexible prophylaxis regimens for the patients: (1) reducing the number of infusion in patients with a high treatment burden, (2) reducing the rFVIII IU consumption keeping the same treatment frequency, and (3) increasing the protection against bleeding using the same treatment regimen (dose and frequency) used with the previous rFVIII generation or a slightly higher dose. However, the improvement associated with BAY 81-8973 and reported in our study could be related to other aspects. Patients could had a different adherence to 2nd- or 3rd-generation SHL rFVIII compared with BAY 81-8973; unfortunately, we lack these types of data due to the retrospective nature of the study. Further, change from an old to a new product can make the patients more caution in their daily life due to the lack of experience with the new treatment. Finally, some of patients involved in our study experienced a significant amount of bleeding in the year before the treatment switch, with a possible impact on their life style and activities during the first year with the prophylaxis with BAY 81-8973.

An improvement in haemophilia management, in terms of effectiveness, resource consumption, and/or treatment burden has been also associated with pharmacokinetics driven (PK-driven) prophylaxis and the use of extended half-life (EHL) rFVIII concentrates [14,15,16,17,18,19]. Recently, many studies using the same approach applied in our evaluation have assessed the impact of EHL rFVIII products in patients previously treated in prophylaxis with standard half-life [14,18,19]. EHL rFVIII products are made using technology designed to extend rFVIII half-life and reported an extended half-life ratio measured in a PK comparator crossover study [20]. The first evidence derived from these studies suggested a reduction of treatment burden and an improving in clinical outcomes. However, this studied included patients treated with 2nd- and 3rd-generation standard half-life products, with no patients treated with BAY 81-8973 before the switch to EHL rFVIII products.

The new treatment approaches and rFVIII products that have become available in the last years in the haemophilia field have afforded the possibility to improve its management, reducing the treatment burden, FVIII consumption and providing similar or lower bleeding rates compared with previous generations of rFVIII concentrates and standard dosing regimens (every-other-day or two-times-a-week infusions). BAY 81-8973 is a SHL product based on a fixed dosage (not PK-driven), though it has shown the possibility to improve patient treatments and outcomes thanks to the better PK profile vs. 2nd or 3rd generation SHL rFVIII products. The value of the treatment is supported by the significant positive impact on ABR and IU consumption showed in our study. The reduction of ABR reported by BAY 81-8973 could also have long-term positive clinical and economic implications considering the reduction of bleeding and the relative long-term effect on joint damage, haemophilic arthropathy, disability onset, and need of orthopedic surgery. However, this information must be assessed in a long-term study.

The study reported some strengths and limitations that must be discussed. All patients were managed in five Haemophilia Treatment Centers that provide comprehensive haemophilia care with multidisciplinary teams, resulting in detailed medical records and the habit to manage new therapeutic options. Expert haematologists who worked in the Centers reviewed the records of all patients treated with BAY 81-8973 to guarantee the quality of the data. However, the analysis was limited by the retrospective nature of the study and the extent and quality of the information available within the records in the Centers and from a small number of patients. The majority of patients were treated with 2nd-generation rFVIII before to switch to BAY 81-8973, so our results could not be generalized to all 2nd- and 3rd-generation rFVIII products. All patients were 12 years old or older and our results need to be confirmed in children <12 years. Based on the sample size and the data available, we were not able to perform more sophisticated statistical analyses (e.g., estimation of minimal detectable change, analyses based on independent variables—age, region, etc.). The analyses performed in the study did not allow results to be generalized, and further studies with a larger patient cohort are required to confirm the first indication of our study and investigate the data with more specific analyses. Finally, the availability of new EHL rFVIII concentrates and non-replacement therapy requires further assessment to understand their relative value compared with BAY 81-8973.

## 5. Conclusions

Our study provided new evidence to understand the real-world impact of prophylaxis with BAY 81-8973, a non-modified SHL rFVIII product with indication also for a two-times-a-week regimen. The use of BAY 81-8973 in patients enrolled in our study resulted in an improvement in ABR and a reduction in both frequency of administration and overall FVIII consumption, highlighting the possibility of a more flexible dosing regimen. An approach that reduces the number of infusions, such as that shown for BAY 81-8973, could lead to an improvement in the patient quality of life, as well as inducing potential savings for the National Health System. Besides, for many patients, a change of product could represent a good opportunity for discussion about revision and adaption of treatment to better suit their individual needs. The impact of these advantages in clinical practice must be confirmed in a larger and heterogeneous patient population, including costs and comparisons with new EHL rFVIII and non-replacement therapies.

## Figures and Tables

**Figure 1 jcm-11-03434-f001:**
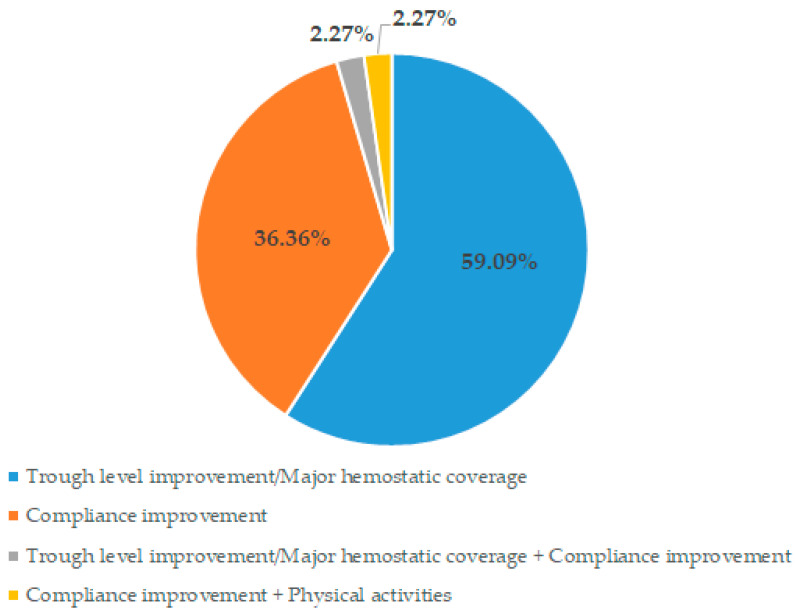
Reasons for switching to BAY 81-8973.

**Figure 2 jcm-11-03434-f002:**
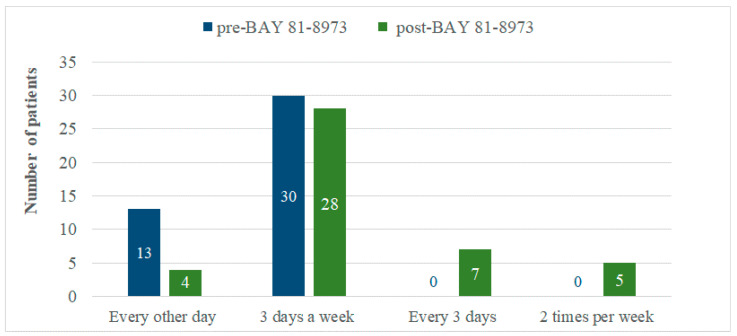
Difference of frequency of administration reported in pre and post BAY 81-8973.

**Table 1 jcm-11-03434-t001:** Baseline Demographic and Clinical Characteristics.

Patients	N (%)
N° of patients	44
Sex: Male, N (%)	44 (100.00%
Age, Median (Range)	35.93 (12–72)
**Clinical characteristics**	
Haemophilia severity, N (%)	
Severe	42 (95.45%)
Moderate	1 (2.27%)
Mild	1 (2.27%)
Patient with target joint, N (%)	25 (56.82%)
Prophylaxis regimen, N (%)	44 (100.0%)
Prophylaxis duration (year), Median (Range)	9.0 (1–20)
Products used in the 12 months before BAY 81-8973, N (%)	
Kogenate FS	33 (75.00%)
Helixate Nexgen	8 (18.18%)
Advate	1 (2.27%)
Refacto AF	1 (2.27%)
More than one product	1 (2.27%)

**Table 2 jcm-11-03434-t002:** Clinical outcomes.

	One Year Pre-BAY 81-8973	One Year Post-BAY 81-8973	Δ (%)	*p*-Value *
Total Bleeds, N	78	10	**−68**	
Joint Bleeds, N (%)	58/73 ^ (79.45%)	7/10 (70.00%)		
Patients with zero bleeds, N (%)	24 (54.55%)	37 (84.09%)	**13** **(+54.2%)**	0.003
**All patients**				
ABR, Mean (± SD)	1.76 (±4.15)	0.23 (±0.60)	**−1.53** **(−87.0%)**	0.015
ABR, Median (range)	0.00 (0–25)	0.00 (0-3)	**0**	
**Patients with ≥1 bleed**				
ABR, Mean (±SD)	3.90 (±6.07)	1.43 (±0.79)	**−2.47** **(63.3%)**	<0.001
ABR, Median (range)	2.00 (1–25)	1.00 (1–3)	**−1.0** **(−50.0%)**	

ABR = Annual bleeding rate. * *p*-value estimated with paired Wilcoxon Signed-Rank Test. ^ One patient reported five bleeds without specifying the site.

**Table 3 jcm-11-03434-t003:** Prophylaxis regimen and FVIII consumption.

	One Year Pre-BAY 81-8973	One Year Post-BAY 81-8973	Δ (%)	*p*-Value *
**Prophylaxis**				
IU/Kg/Infusion Mean (±SD)	30.66 (±5.03)	31.65 (±4.96)	+0.99 (+3.20%)	
N Infusion per week, Mean (±SD)	3.04 (±0.42)	2.83 (±0.43)	−0.21 (−7.00%)	
**Annual prophylaxis consumption per patient °, IU**	**341,319.36**	**327,706.47**	**−13,612.89** **(−4.00%)**	**0.117**
**Treatment of bleeds**				
ABR, Mean (±SD)	1.76 (±4.15)	0.23 (±0.60)	**−1.53** **(−87.0%)**	
IU/Kg/Infusion, Mean (±SD)	36.95 (±4.15)	36.14 (±11.54)	−0.81 (−2.20%)	
N ° of Infusion, Mean (±SD)	3.00 (±1.84)	3.14 (±3.53)	0.14 (4.70%)	
**Annual bleeding consumption per patient °, IU**	**13,746.29**	**1817.47**	**−11,928.82** **(−86.70%)**	**<0.001**
**Total Consumption per patients °, IU**	**355,065.65**	**329,523.94**	**−25,541.71** **(−7.20%)**	**0.046**

° Estimated on average weight of 70.4 kg. * *p*-value estimated with paired Wilcoxon Signed-Rank Test.

## Data Availability

The data presented in this study are available on request from the corresponding author. The data are not publicly available due to privacy reason.

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
