# Peer review of "Real-World Clinical Outcomes and Replacement Factor VIII Consumption in Patients with Haemophilia A in Italy: A Comparison between Prophylaxis Pre and Post Octocog Alfa (BAY 81-8973)"

_jcm, 2022, doi:10.3390/jcm11123434_

Round 1

Reviewer 1 Report

The authors have carried out an interesting retrospective analysis of the efficacy of one of the pioneering drugs in the approach to hemophilia less than a decade ago. However, to make it easier for readers less familiar with hemophilia to understand, and to be able to generalize the results, some changes to the text are required:

Introduction

  • At the beginning it is indicated that hemophilia affects the perception of quality of life of these patients. This statement should be made after relating the coping with the disease and the joint sequelae with this perception deficit.
  • It is important, for readers not familiar with hemophilia, to explain the types of hemophilia based on coagulation factor deficiency, and the percentage of coagulation factor in the blood.
  • The process of joint damage characteristic of hemophilia (hemophilic arthropathy) is not clearly explained, briefly and concisely. The final objective of the study is to observe the frequency of bleeding, which causes arthropathy. therefore, it should be stated that it is desired to avoid in the long term.
  • There is talk of cerebral hemorrhages (5% of hemorrhages in these patients). The low prevalence of these should be noted. Otherwise the reader can interpret that they are frequent in hemophilia.
  • When it is said that the prophylaxis protection is not optimal (line 53) it must be justified why it is not.

Methods

  • Study design not stated.
  • Exclusion criteria are included that are the opposite of inclusion criteria. It should not be like that. Both criteria must be independent and complementary, not the opposite.
  • The patient recruitment process is not explained, nor the dates of inclusion in the study.
  • The sample size necessary to obtain a representative sample has not been calculated.
  • Were the joint bleeds confirmed by patient information, physical examination by the doctor, ultrasound...? It should follow the same diagnostic criteria, to avoid confusion with acute synovial hypertrophies, chronic referred pain that the patient/doctor confuses with a hemarthrosis, etc.
  • Statistical Analysis: Analysis based on real time (months/days) with Kovaltry, prior drug type, dosage, treatment schedule has not been performed. A basic statistical analysis has been done that cannot allow the results to be generalized. I recommend calculating the minimum detectable change and analysis based on the independent variables (age, type of hemophilia, prior primary/secondary prophylaxis, dosage, region, etc.)

Discussion

  • The discussion is very general and does not emphasize the reason for the changes with rFVIII. I recommend that the authors explain the advantages of the active ingredient compared to other rFVIII products.
  • The generalization of the results requires more powerful statistical analyzes with independent variables. With this, the authors will be able to write a much more direct and effective discussion to convince the reader of their purpose.
  • Study limitations should be more self-critical.

Tables and figures

  • I recommend using a single statistic for central tendency (mean or median) and dispersion (standard deviation or interquartile range) in the descriptive analysis.

Author Response

Reviewer 1

The authors have carried out an interesting retrospective analysis of the efficacy of one of the pioneering drugs in the approach to hemophilia less than a decade ago. However, to make it easier for readers less familiar with hemophilia to understand, and to be able to generalize the results, some changes to the text are required.

We thank the reviewer for the comments to our manuscript. In the new version of the manuscript we carefully considered all raised points. 

Our point-by-point responses are provided below. 

Introduction

  • At the beginning it is indicated that hemophilia affects the perception of quality of life of these patients. This statement should be made after relating the coping with the disease and the joint sequelae with this perception deficit.

We thank the reviewer for the comment, we have changed this part of the introduction.

  • It is important, for readers not familiar with hemophilia, to explain the types of hemophilia based on coagulation factor deficiency, and the percentage of coagulation factor in the blood.

We thank the reviewer for the comment, we have better explained this aspects.

  • The process of joint damage characteristic of hemophilia (hemophilic arthropathy) is not clearly explained, briefly and concisely. The final objective of the study is to observe the frequency of bleeding, which causes arthropathy. therefore, it should be stated that it is desired to avoid in the long term.

We thank the reviewer for the comment, we have included this aspects in the introduction.

  • There is talk of cerebral hemorrhages (5% of hemorrhages in these patients). The low prevalence of these should be noted. Otherwise the reader can interpret that they are frequent in hemophilia.

We thank the reviewer for the comment, we have better explained this aspects.

  • When it is said that the prophylaxis protection is not optimal (line 53) it must be justified why it is not.

We thank the reviewer for the comment, we have better explained the issue related to the prophylaxis.

Methods

  • Study design not stated.

We have specified in the method section that the study is a retrospective, observational, multicenter cohort study.

  • Exclusion criteria are included that are the opposite of inclusion criteria. It should not be like that. Both criteria must be independent and complementary, not the opposite.

We thank the reviewer for the comment, we have better specified the exclusion criteria.

  • The patient recruitment process is not explained, nor the dates of inclusion in the study.

We thank the reviewer for the comment, we have better specified the patients enrolment.

  • The sample size necessary to obtain a representative sample has not been calculated.

Due to the observational nature of the study and considering that haemophilia is a rare disease, a specific sample size calculation was not performed. The study included all patients that satisfy the inclusion and exclusion criteria in the centers participating the study.

Mingot-Castellano ME 36 patients

Dunn AL 15 patients

Tagliaferri A 18 patients

  • Were the joint bleeds confirmed by patient information, physical examination by the doctor, ultrasound...? It should follow the same diagnostic criteria, to avoid confusion with acute synovial hypertrophies, chronic referred pain that the patient/doctor confuses with a hemarthrosis, etc.

We thank the reviewer for the comment; we have better specified in the method section how bleeding was defined. The bleeding is by definition of haemophilia comunity a patients reported outcomes (Manco-Johnson MJ, Warren BB, Buckner TW, Funk SM, Wang M. Outcome measures in Haemophilia: Beyond ABR (Annualized Bleeding Rate). Haemophilia. 2021 Feb;27 Suppl 3:87-95; Pierce G, Ragni M, Van den Berg H, Weill A, O'Mahony B, Skinner M, et al. Establishing the appropriate primary endpoint in haemophilia gene therapy pivotal studies. Haemophilia. 2017;23:643–44.), and in the clinical practice it is normally reported by patents to the clinicians using a patient diary or reporting during the control visit. This approach was the same in all centers involved.

  • Statistical Analysis: Analysis based on real time (months/days) with Kovaltry, prior drug type, dosage, treatment schedule has not been performed. A basic statistical analysis has been done that cannot allow the results to be generalized. I recommend calculating the minimum detectable change and analysis based on the independent variables (age, type of hemophilia, prior primary/secondary prophylaxis, dosage, region, etc.)

We thank the reviewer for the comments, however based on the number of patients involved in the study an analysis based on independent variables has not enough data to be performed. Further, also the drug type used before is the same in the 93.2% of patients.

We have performed a basic statistical analysis to provide a preliminary information on the possible impact of product with extended frequency of infusion as Kovaltry on patients’ outcomes and FVII consumption. We have reported the limits of the analyses in the discussion; further, we have suggested the need of future studies with the power to assess the impact of these products using the analyses suggested by the reviewer.

Discussion

  • The discussion is very general and does not emphasize the reason for the changes with rFVIII. I recommend that the authors explain the advantages of the active ingredient compared to other rFVIII products.

We have underlined the advantage of the new product compare to the previous generation.

  • The generalization of the results requires more powerful statistical analyzes with independent variables. With this, the authors will be able to write a much more direct and effective discussion to convince the reader of their purpose. Study limitations should be more self-critical.

We have reported in the limit section the issues associate to the analysis performed and the limit in the results generalization.

Tables and figures

  • I recommend using a single statistic for central tendency (mean or median) and dispersion (standard deviation or interquartile range) in the descriptive analysis

We thank the reviewer for the comment; we have kept only one single statistic for central tendency and dispersion in all tables except for the ABR reported in the Table 2. This choice is made on the need of providing statistics that can be compare with other studies. Unfortunately, there is heterogeneity in reporting ABR, some using mean and other median, and this create issues in comparing the data between studies. Based on the situation we prefer to reporting ABR as mean and median.

Reviewer 2 Report

1.The manuscript was interesting.

2.The topic was original. However, I wonder whether to put the tradename of Kovaltry at the title is appropriate. Should the word “Kovaltry” be deleted from the title and manuscript? It may be stated in the Materials and Methods. Then unmodified and full-length rFVIII should be used.

3.Introduction: line 67-69 of the introduction with reference 12 published since 1992. More update references should be cited

4.The manuscript added information to the subject area of standard half-life rFVIII.

5.The manuscript was well-written and understandable. However, the reason of favorable outcome among studied patient receiving Kovaltry was not clearly explained. The advantage of unmodified, full-length standard half-life rFVIII over the previously received second and third generation rFVIII was not clearly stated. Also, no pharmacokinetic study among the studied patients was provided.

6.Results: the dose of Kovaltry was 31.65 IU/kg/infusion while the dose of previously received second and third rFVIII was 30.66 IU/kg/infusion, the slight increase in the dose could not explain the favorable outcome. It should be the formulation of Kovaltry inducing more effectiveness in prevention of bleeding compared to those of previously received second and third generation rFVIII.

7.The authors concluded upon their finding that the administration of Kovaltry twice weekly is effective in decreasing ABR and utilized factor concentrate by the clinical evidence without laboratory support of pharmacokinetics.

Author Response

1.The manuscript was interesting. The topic was original.

Thank you for the opportunity to address the reviewer’s comments, which we have carefully considered when preparing the revised version of the manuscript.

Our point-by-point responses are provided below. 

However, I wonder whether to put the tradename of Kovaltry at the title is appropriate. Should the word “Kovaltry” be deleted from the title and manuscript? It may be stated in the Materials and Methods. Then unmodified and full-length rFVIII should be used.

We thank the reviewer for the comment, we have replace Kovaltry with the reference number used in clinical studies (BAY 81-8973).

2.Introduction: line 67-69 of the introduction with reference 12 published since 1992. More update references should be cited

We thank the reviewer for the comment; we have replace the references with a more recent one.

3.The manuscript added information to the subject area of standard half-life rFVIII.The manuscript was well-written and understandable. However, the reason of favorable outcome among studied patient receiving Kovaltry was not clearly explained. The advantage of unmodified, full-length standard half-life rFVIII over the previously received second and third generation rFVIII was not clearly stated. Also, no pharmacokinetic study among the studied patients was provided.

We thank the reviewer for the comment; we have better explain this aspect in the discussion. We have put tighter our results and TAURUS study results to provide a complete picture of the possible advantages associated to Kovaltry compare to the older SHL-rFVIII. In this way we added also additional information on PK profile.

4.Results: the dose of Kovaltry was 31.65 IU/kg/infusion while the dose of previously received second and third rFVIII was 30.66 IU/kg/infusion, the slight increase in the dose could not explain the favorable outcome. It should be the formulation of Kovaltry inducing more effectiveness in prevention of bleeding compared to those of previously received second and third generation rFVIII.

We thank the reviewer for the comment. We agree that the slightly increase of dose cannot explain the favorable outcome. In the discussion, we reported some possible aspects that could have increase the positive results associated to Kovaltry base do the observational nature of the study.

5.The authors concluded upon their finding that the administration of Kovaltry twice weekly is effective in decreasing ABR and utilized factor concentrate by the clinical evidence without laboratory support of pharmacokinetics.

We thank the reviewer for the comment. We have underline in the discussion that more than 25% of patients reported every 3 days or 2 times per week regimen when treated with Kovaltry® compared to 0% when treated with the previous product. This result confirms what observed in the LEOPOLD studies program, that BAY 81-8973 is able to provide effective prophylaxis with two injections per week in approxi-mately 30% of patients

No pharmacokinetics data were available in the dataset of this study to support the clinical evidence; however, a favorable PK profile of Kovaltry in comparison with other standard half-life rFVIIIs was reported in other study [TAURUS study].
